# The Association of Selected GWAS Reported AD Risk Loci with CSF Biomarker Levels and Cognitive Decline in Slovenian Patients

**DOI:** 10.3390/ijms241612966

**Published:** 2023-08-19

**Authors:** David Vogrinc, Milica Gregorič Kramberger, Andreja Emeršič, Saša Čučnik, Katja Goričar, Vita Dolžan

**Affiliations:** 1Pharmacogenetics Laboratory, Institute of Biochemistry and Molecular Genetics, Faculty of Medicine, University of Ljubljana, 1000 Ljubljana, Slovenia; david.vogrinc@mf.uni-lj.si (D.V.); katja.goricar@mf.uni-lj.si (K.G.); 2Department of Neurology, University Medical Centre Ljubljana, 1000 Ljubljana, Slovenia; milica.kramberger@kclj.si (M.G.K.); andreja.emersic@kclj.si (A.E.); sasa.cucnik@kclj.si (S.Č.); 3Faculty of Medicine, University of Ljubljana, 1000 Ljubljana, Slovenia; 4Department of Neurobiology, Care Sciences and Society, Division of Clinical Geriatrics, Karolinska Institutet, 14152 Huddinge, Sweden; 5Department of Rheumatology, University Medical Centre Ljubljana, 1000 Ljubljana, Slovenia; 6Faculty of Pharmacy, University of Ljubljana, 1000 Ljubljana, Slovenia

**Keywords:** Alzheimer’s disease, mild cognitive impairment, polymorphism, biomarker

## Abstract

Alzheimer’s disease (AD) is the most common neurodegenerative disease, with a complex genetic background. Apart from rare, familial cases, a combination of multiple risk loci contributes to the susceptibility of the disease. Genome-wide association studies (GWAS) have identified numerous AD risk loci. Changes in cerebrospinal fluid (CSF) biomarkers and imaging techniques can detect AD-related brain changes before the onset of clinical symptoms, even in the presence of preclinical mild cognitive impairment. In this study, we aimed to assess the associations between SNPs in well-established GWAS AD risk loci and CSF biomarker levels or cognitive test results in Slovenian patients with cognitive decline. The study included 82 AD patients, 28 MCI patients with pathological CSF biomarker levels and 35 MCI patients with normal CSF biomarker levels. Carriers of at least one polymorphic *TOMM40* rs157581 C allele had lower Aβ_42_ (*p* = 0.033) and higher total tau (*p* = 0.032) and p-tau_181_ levels (*p* = 0.034). Carriers of at least one polymorphic T allele in *SORCS1* rs1358030 had lower total tau (*p* = 0.019), while polymorphic *SORCS1* rs1416406 allele was associated with lower total tau (*p* = 0.013) and p-tau_181_ (*p* = 0.036). In addition, carriers of at least one polymorphic T allele in *BCHE* rs1803274 had lower cognitive test scores (*p* = 0.029). The study findings may contribute to the identification of genetic markers associated with AD and MCI and provide insights into early disease diagnostics.

## 1. Introduction

Alzheimer’s disease (AD) is the most common neurodegenerative disease, affecting a significant part of the worldwide population. It is a leading cause of dementia, that typically occurs in the elderly population, with the vast majority of cases in people aged 65 or older [1]. As the elderly population at increased risk for developing late-onset AD is expected to continue to grow in the future, the disease prevalence will continue to increase, making AD one of the most important health and societal issues. Multiple risk factors contribute to the development of the disease, with age being one of the most important [2]. The appearance of symptoms in the younger population is unusual and is considered early-onset AD [3].

Pathophysiologically, AD manifests with two major disease hallmarks–deposition of amyloid β (Aβ) in neuritic plaques and accumulation of tau protein neurofibrillary tangles [4]. The induced brain changes can be measured with cerebrospinal fluid (CSF) biomarkers (Aβ_42_, Aβ_40_, p-tau_181_) and modern imaging approaches (PET scan). Although AD generally manifests in the elderly population, initial changes in CSF biomarker levels can be observed years before the onset of clinical symptoms [5,6]. The protective mechanisms of the brain are still sufficient at this preclinical stage, to prevent further impairment and memory loss [1]. Mild cognitive impairment (MCI) is the main feature of the pre-dementia stage of the disease, with changes in AD biomarkers and subtle cognitive impairments [7]. These cognitive problems are noticeable but do not affect daily activities. MCI may eventually progress to AD. It is estimated that up to one-third of MCI patients develop dementia due to AD within five years [8]. Still, some individuals with MCI do not progress to AD [1]. It is clinically relevant to identify individuals at higher risk for developing AD; thus, the search for predictive AD biomarkers is of great importance in current research. 

Similar to other common chronic diseases, a combination of multiple factors contributes to the development of AD. A small proportion of AD cases show a familial, highly heritable form of AD [4]. Mutations in three genes—amyloid precursor protein (*APP*), presenilin-1 (*PSEN1*), and presenilin-2 (*PSEN2*)—were found in familial AD with earlier onset of symptoms [3,9]. However, there are no common causative genes for the sporadic form of the disease. Numerous genome-wide association studies (GWAS) have identified different AD risk loci [10,11,12,13]. Of the many genes that increase AD risk, *APOE* has the strongest impact on late-onset AD. Two common *APOE* polymorphisms, rs429358 (*p*.Cys112 Arg) and rs7412 (*p*.Arg158 Cys), define a combination of alleles (ε2, ε3, and ε4) that affect AD risk [14]. The risk for AD is 2-3-fold higher in carriers of one *APOE* ε4 allele and about 12-fold higher in those with two *APOE* ε4 alleles [15,16]. On the other hand, the *APOE* ε2 allele is protective [17]. In addition to *APOE*, many other genes involved in different molecular pathways contribute to susceptibility to develop AD.

In a previously published systematic review, we used GWAS data to identify key pathways of AD pathogenesis: cellular processes, metabolic processes, biological regulation, localization, transport, regulation of cellular processes, and neurological system processes [18]. Among genes involved in localization pathways, *TOMM40* and *SORCS1* were frequently reported as potential AD biomarkers. *TOMM40* encodes the subunit of outer mitochondrial membrane translocase, a channel-forming pore for protein uptake in mitochondria [19]. *SORCS1* is a member of the Vps10 p family of sorting receptors, important in APP processing [20]. Serine esterase BCHE is important in neurotransmitter activation and has been found to be enriched in senile plaques of AD brains [21,22]. Angiotensin-converting enzyme, encoded by the *ACE* gene is primarily known as a vasoconstrictor; however, its role in Aβ degradation has also been reported [23]. IL6 R is the receptor for IL6, a cytokine important for neuronal cell growth and differentiation. The genetic variability of *IL6 R* was previously studied in association with elevated IL6 activity in AD brain [24].

The search for robust and non-invasive biomarkers of early AD detection is still ongoing. Therefore, the aim of our study was to investigate the association of genetic variability in genes, previously identified in GWAS studies with AD biomarker levels and cognitive decline in Slovenian patients with AD and MCI.

## 2. Results

### 2.1. Patients’ Characteristic

Our study included 145 patients with cognitive impairment: 82 AD patients, 28 MCI (AD), and 35 MCI (NOT AD). Clinical characteristics of all patients with cognitive impairment and of each separate group (AD, MCI (AD) and MCI (NOT AD)) are summarized in Table 1. AD patients were significantly older compared to patients with MCI (AD) and MCI (NOT AD) (*p* = 0.039). More female subjects were included in both AD and MCI (AD) groups (*p* = 0.008). Significant differences in all CSF biomarker levels (Aβ_42_, Aβ_42/40_, total tau and p-tau_181_) were observed between groups (all *p* < 0.001). AD patients also achieved significantly lower MMSE scores (*p* < 0.001).

Genotype frequencies of all nine investigated SNPs in *SORCS1*, *BCHE*, *TOMM40*, *ACE* and *IL6 R* genes in the entire cohort are presented in Appendix A.

### 2.2. Association of Investigated SNPs with Cognitive Impairment and AD Susceptibility

The genotype frequency distribution of investigated polymorphisms in different patient groups with cognitive impairment is presented in Table 2. *APOE* rs429358 polymorphic C allele was more frequent in AD and MCI (AD) (*p* = 0.004). Normal *TOMM40* rs157581 T allele was more frequent in MCI (not AD) group and the difference was statistically significant in both additive and dominant models (*p* = 0.005 and *p* = 0.001, respectively). Furthermore, at least one polymorphic *TOMM40* rs2075650 G allele was more frequent in the AD and MCI (AD) group, compared to MCI (not AD); however, the nominal difference was only observed in the dominant model (*p* = 0.025).

### 2.3. Association of Investigated SNPs with CSF Biomarker Levels and MMSE

Among investigated SNPs, associations of *TOMM40* rs157581 with different CSF biomarkers were observed among all patients with cognitive impairment (Table 3). Carriers of at least one polymorphic C allele had lower Aβ_42_ (*p* = 0.033) and higher total tau (*p* = 0.032) and p-tau (*p* = 0.034) levels, respectively. However, the associations did not remain significant when only AD patients were included in the analysis (Appendix A). On the other hand, significant associations of *SORCS1* polymorphisms with tau biomarkers were observed in the AD group. Carriers of at least one polymorphic *SORCS1* rs1358030 T allele had lower total tau (*p* = 0.019). Carriers of one polymorphic C allele in *SORCS1* rs1416406 had lower total tau (*p* = 0.013) and p-tau (*p* = 0.036) levels (Figure 1). The association did not remain significant in the dominant model.

Nominally significant associations of *BCHE* rs1803274 with the MMSE cognitive test scores were observed in the entire group of patients with cognitive impairment. Carriers of at least one polymorphic T allele had lower test scores (*p* = 0.029) (Figure 2). No significant or nominally significant associations with MMSE were observed for other investigated SNPs (Appendix A).

## 3. Discussion

We investigated the association of selected SNPs from GWAS-identified AD risk genes with CSF biomarker levels and cognitive test results in patients with cognitive impairment. *TOMM40* and *SORCS1* were associated with both amyloid and tau CSF biomarkers, while *BCHE* was associated with cognitive decline.

Numerous GWAS and case–control studies identified these genes as potential AD risk loci [25,26,28,29,30,31]. Among investigated SNPs, both *TOMM40* rs157581 and rs2075650 were previously associated with increased AD susceptibility in European [32,33] and Asian [27,34,35] populations. Similarly, increased AD risk was observed for *BCHE* rs1803274 in European [36] and *ACE* rs1800764 in the Asian population [37]. Furthermore, the genetic variance within these genes was also associated with Aβ and tau levels in CSF [38,39,40,41] and blood plasma [42,43]; therefore, our study focused on common functional polymorphisms in established AD risk loci. In our study, lower CSF Aβ_42_ and higher total tau and p-tau levels have been observed in carriers of the polymorphic C allele in *TOMM40* rs157581. As previously shown, TOMM40 genetic variability is associated with CSF Aβ and tau levels [38,44]. Polymorphic alleles in both *SORCS1* SNPs have been associated with lower total tau, while carriers of *SORCS1* rs1416406 C allele additionally had lower p-tau. In concordance with the literature, the APOE effect on amyloid and tau pathology was also confirmed.

Among investigated genes, the *TOMM40* was the only one associated with both Aβ and tau CSF biomarkers. On a pathophysiological level, *TOMM40* was associated with APP accumulation in the mitochondrion. APP is the precursor molecule for processing of Aβ proteins of various lengths [45] and APP mutations result in overproduction and aggregation of Aβ_42_ [4,46]. By entering and obstructing the TOMM40 pore, APP induces mechanisms for mitochondrial dysfunction [47]. Furthermore, *TOMM40* impacts AD-vulnerable brain areas by downstream apoptotic processes that forego extracellular Aβ aggregation. Considering those pieces of evidence, *TOMM40* is an important gene presumably contributing to AD-related mitochondria dysfunction [48]. Its genomic location, adjacent to the *APOE* region gene on chromosome 19, raised interest in the assessment of *TOMM40* genetic variability on AD susceptibility. Both *TOMM40* rs157581 and rs2075650 were associated with AD risk in GWAS and meta-analyses [25,28,30,49,50]. Our present study shows the effect of *TOMM40* rs157581 on CSF biomarkers, thus further supporting the importance of this gene in the development of AD. However, CSF Aβ_42_, p-tau_181_/Aβ_42_ and total-tau/Aβ_42_ as quantitative traits in GWAS were previously associated with rs2075650 [38]. A similar effect was observed in another study [39], where *TOMM40* rs2075650 showed a genome-wide association with CSF tau and Aβ_42_. In concordance with our results, the genome-wide association of *TOMM40* rs2075650 with lower Aβ_42_ levels in CSF of AD patients was also reported [44]. Since the close proximity to *APOE*, the potentially combined effect of *TOMM40-APOE* on AD pathology was also addressed. The polygenic profile of *APOE-TOMM40-APOC1* was associated with increased CSF tau levels, suggesting the important role of *TOMM40* in the modulation of the *APOE* ε4 effect in AD [51]. As previously reported, our study confirmed both decreased Aβ_42_ and increased tau CSF levels.

We have also observed the association between both *SORCS1* polymorphisms and tau CSF levels. *SORCS1* is another gene important in APP processing. Since SorCS1 is prominently expressed in the nervous system [52], it might affect AD pathophysiology. Overexpression of SorCS1 might lead to lower Aβ levels through reduced γ-secretase activity [53]. Aβ is derived by the proteolytic cleavage of APP by a successive β-secretase cleavage at the N-terminus, followed by γ-secretase cleavage of the membrane-bound C-terminal site [4,54]. *SORCS1* was proposed as an AD risk locus in several GWAS [53,55,56]. To the best of our knowledge, no association studies on CSF biomarkers were conducted on AD patients. The observed associations of genetic variability in *SORCS1* with tau pathology in our study are, thus, novel findings that can further contribute to the understanding of the effect of APP processing on tau aggregation.

Apart from CSF biomarkers, biomarkers related to cognitive decline can also serve as a marker of neurodegeneration. The observed lower MMSE scores associated with *BCHE* rs1803274 highlight the potential of serine esterase in AD. Butyrylcholinesterase (BCHE) as a serine esterase is involved in organophosphate ester hydrolysis [21]. It is important in neurotransmitter activation and elevated BCHE activity was observed in the brain affected by amyloid plagues [22,57]. Furthermore, a link between BCHE and the accumulation of tau protein in neurofibrillary tangles was found in AD brain [58]. In terms of genetic variability, a significant association of *BCHE* rs509208 with cortical Aβ in AD subjects was found in GWAS [40]. The effect of the *BCHE* rs1803274 genotype on impaired BCHE activity in brain tissue and CSF was found [36,59]. What is more, the effect of *BCHE* rs1803274 on cognitive decline, measured with MMSE in combination with donepezil treatment in MCI treatment was evaluated [60]. Similar to our findings, a faster MMSE decline was associated with a polymorphic allele. Our results thus confirm the previously found negative effect of BCHE on cognitive decline. Although they were previously associated with AD risk in GWAS, polymorphisms in *ACE* and *IL6 R* did not reach significant or nominally significant associations with CSF biomarker levels and MMSE scores.

Our study has some limitations. The sample size of different pathologies was relatively small and some clinical parameters, especially cognitive test scores, were not available for all patients. We are aware, that our study cannot compare with larger consortium-based approaches in GWAS studies, for the assessment of disease risk loci. However, the detailed patient-related data enabled us to focus assessment of the genetic variability of selected genes in relation to CSF biomarkers and cognitive tests in addition to evaluating these SNPs as potential risk factors. The smaller sample size is therefore partly due to the fact that the patients were included during their lumbar punction appointment to assess CSF biomarkers. Although it is difficult to detect the contributions of many factors in a smaller sample size study, a similar effect can occur due to phenotypic heterogeneity in larger studies as well. We also accounted for multiple comparisons in the statistical analysis. For a more thorough evaluation of observed associations, an independent study with a larger sample could be conducted. On the other hand, our study had several strengths. All the patients were recruited from the same department and evaluated according to the same protocol. We comprehensively assessed the simultaneous influence of several clinical and genetic parameters on AD risk and pathology. We were the first to assess the genetic variability in some of the GWAS-identified AD risk loci in Slovenian patients. Furthermore, only a few studies focused on the association of selected genes with CSF biomarkers. We believe our work might serve as a valuable addition to the field of translation of AD genetic risk to biomarker levels in patients.

## 4. Materials and Methods

### 4.1. Subjects

Our study included patients with cognitive impairment as they had appointments for clinical evaluation and lumbar puncture at the Department of Neurology, University Medical Centre Ljubljana, Slovenia, between June 2019 and December 2022. Inclusion criteria were age above 55 and diagnosis of AD or MCI. Patients with comorbidities significantly affecting cognitive performance and dementia due to diseases other than AD were excluded from the study. A structured interview with patients and their caregivers was performed to obtain demographic and clinical data. Additional information was obtained from medical records.

The study protocol was approved by the National Medical Ethics Committee of the Republic of Slovenia (0120–523/2017–4) and all the subjects provided written informed consent in accordance with the Declaration of Helsinki.

### 4.2. Assessment

Cognitive impairment was diagnosed through a standardized clinical evaluation and an assessment of patients’ cognitive decline history. The Mini-mental State Examination (MMSE) was administered to screen for cognitive deficit [61]. A thorough diagnostic work-up was conducted, combining structural brain imaging, blood laboratory tests, neuropsychological assessment and CSF AD biomarker testing. Following a consensus meeting with clinicians and neuropsychologists, patients received a cognitive impairment diagnosis based on the DSM V criteria [62].

The patients were categorized into three groups, AD, MCI (AD) and MCI (NOT AD), based on CSF biomarker levels, dementia criteria and Winblad & Peterson MCI diagnostic criteria [63], as previously described [64]. Locally validated biomarker cut-off levels were applied for Aβ_42_ (>570 pg/mL), Aβ_42_/_40_ (>0.07), p-tau_181_ (<60 pg/mL) and total tau (<400 pg/mL), respectively. Patients with elevated total, p-tau_181_ and reduced Aβ_42_ and/or Aβ_42_/_40_ levels and with impaired daily activities were defined as the AD group. Patients with MCI and AD CSF biomarker profiles and normal daily functioning were included in MCI (AD) group. Patients with normal biomarker levels and MCI, having preserved daily functioning were assigned to the MCI (NOT AD) group.

### 4.3. Cerebrospinal Fluid Analysis

CSF was obtained via lumbar puncture between the L3/L4 and L4/L5 intervertebral space using a 25 gauge needle and collected in polypropylene tubes (Sarstedt AG & Co., Nümbrecht, Germany) (Figure 3). Biomarker analysis was performed at the Laboratory for CSF Diagnostics, Department of Neurology, University Medical Centre Ljubljana, Slovenia. The levels of Aβ_42_, Aβ_40_, p-tau_181_ and total tau were measured as previously described [64]. CSF biomarker analyses were performed according to manufacturers’ instructions using Innotest (Fujirebio Europe, Gent, Belgium) immunoassays with intra-assay variability < 5%. Between-assay coefficients and locally validated biomarker cut-off levels are listed elsewhere [65].

### 4.4. Genotyping

Genomic DNA was isolated using the E.Z.N.A.^®^ SQ Blood DNA Kit II (Omega Bio-tek, Inc., Norcross, GA, USA) from peripheral venous blood samples following the manufacturer’s protocol. Genotyping was performed for single-nucleotide polymorphisms (SNPs) in 5 genes previously associated with AD in different GWAS studies: *SORCS1*, *BCHE*, *TOMM40*, *ACE*, and *IL6 R* (Figure 3). Polymorphism was selected among the previously identified GWAS genes, based on the data available from the published literature and their function prediction. Only potentially functional SNPs with a minor allele frequency of at least 0.05 were selected. For *SORCS1* and *BCHE*, we choose different SNPs than those that were identified in the GWAS approach, due to the lack of potential functionality. In total, nine SNPs were included in the analysis (Appendix A).

All of the selected SNPs were genotyped with competitive allele-specific PCR (KASP assays, LGC Biosearch Technologies, Hoddesdon, UK), according to the manufacturer’s instructions.

Additionally, *APOE* rs7412 and rs429358 were genotyped for the assessment of APOE4 status using real-time PCR-based Taqman assay (Applied Biosystems, Foster City, CA, USA). Ten percent of samples were genotyped in duplicate as quality control and all the results were concordant.

### 4.5. Statistical Analysis

We used median and interquartile range (25–75%) to describe continuous variables, and frequencies to describe categorical variables. The interquartile range was determined using weighted averages if more than two samples were included in the group and using Tukey’s hinges if only two samples were included in the group. Fisher’s exact test or Kruskal–Wallis test was used to compare patients’ characteristics and genotype frequencies between groups. The agreement of genotype frequencies with Hardy–Weinberg equilibrium (HWE) was assessed using the Chi-squared test. Both dominant and additive genetic models were used in the analysis. Mann–Whitney test or Kruskal–Wallis test with post hoc Bonferroni corrections for pairwise comparisons were used to evaluate the association of SNPs with MMSE and CSF biomarker levels. Bonferroni correction was used to account for multiple comparisons. As nine SNPs were included in the final analysis, the significance threshold was set to 0.0056, and *p*-values below 0.0056 were considered statistically significant, while *p*-values between 0.0056 and 0.050 were considered nominally significant. IBM SPSS Statistics version 27.0 (IBM Corporation, Armonk, NY, USA) was used for all analyses. All tests were two-sided and the level of significance was set at 0.05. Figures were prepared using GraphPad Prism version 9 (GraphPad Software, LLC., San Diego, CA, USA).

Based on the sample size, this study had 80% power to detect differences in CSF Aβ_42_ levels between 167 and 205 pg/ml for polymorphisms with minor allele frequencies between 0.20 and 0.40. Power calculation was performed using the PS power and sample size calculations, version 3.1.6 [66].

## 5. Conclusions

Analysis of genetic variability in GWAS identified AD risk and showed associations with CSF AD biomarker levels and cognitive decline in Slovenian patients. We associated *TOMM40* and *SORCS1* with both amyloid and tau pathologies, whereas *BCHE* was associated with cognitive decline. Our findings support previously published results and also propose some of the new associations with common polymorphisms of AD risk loci involved in various aspects of cellular signaling and localization.

## Figures and Tables

**Figure 1 ijms-24-12966-f001:**
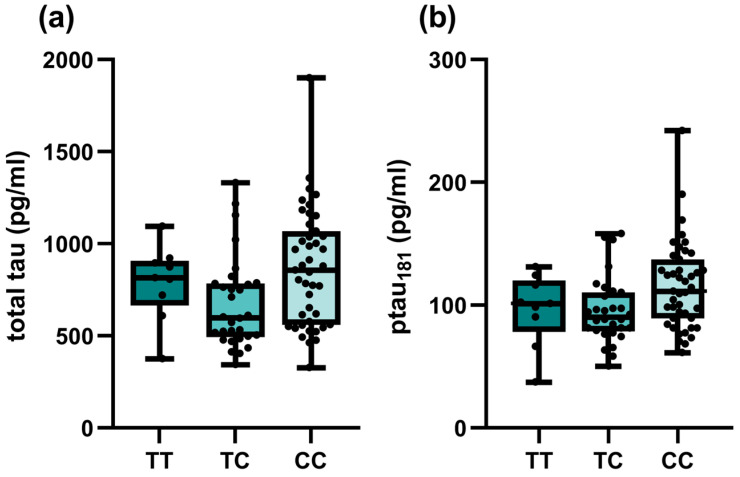
Associations of *SORCS1* rs1416406 genotypes of AD patients and CSF biomarker levels: (**a**) total tau; (**b**) p-tau_181_.

**Figure 2 ijms-24-12966-f002:**
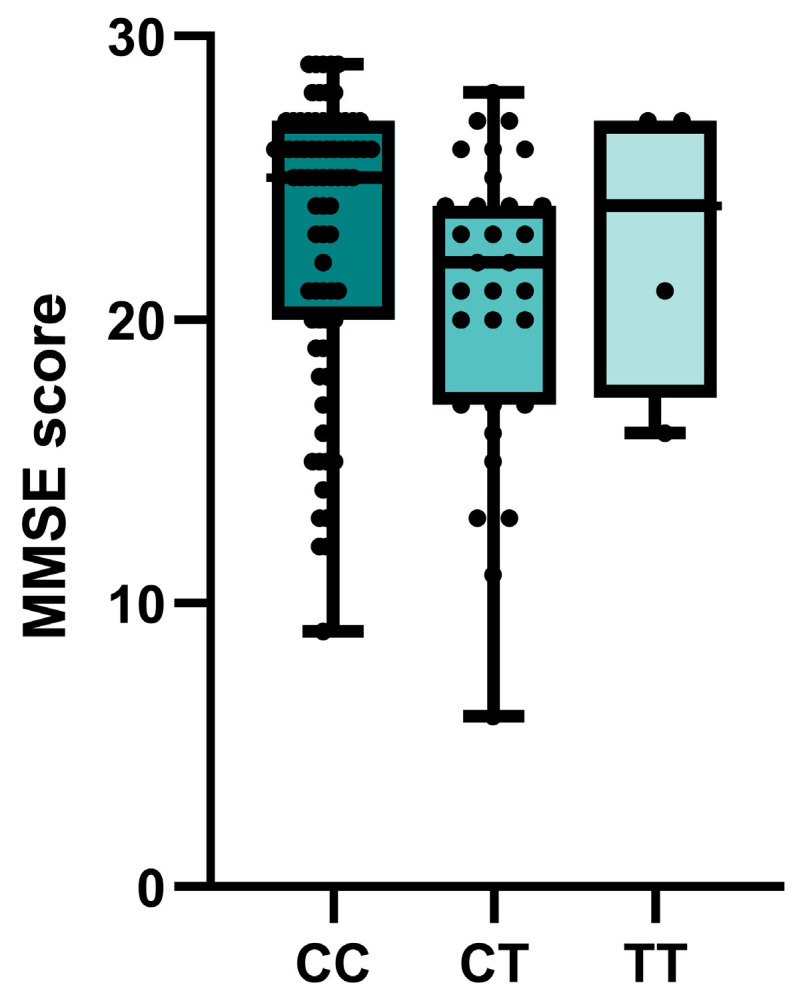
Associations of *BCHE* rs1803274 genotypes of patients with cognitive impairment and MMSE score.

**Figure 3 ijms-24-12966-f003:**
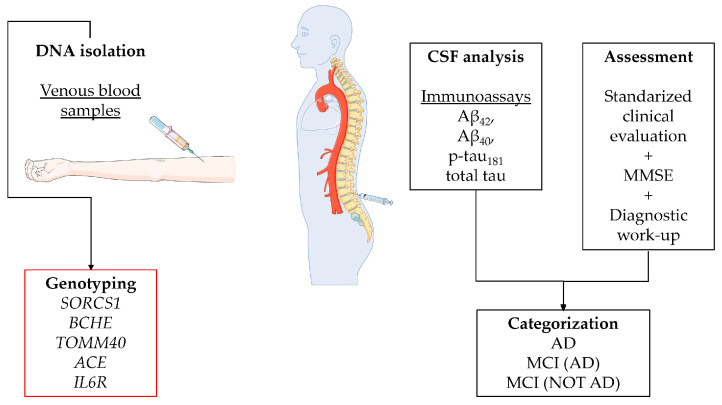
Workflow of methods: DNA isolation and genotyping, CSF analysis and assessment. The Figure was partly generated using Servier Medical Art, provided by Servier, licensed under a Creative Commons Attribution 3.0 unported license (https://creativecommons.org/licenses/by/3.0/ 20 July 2023).

**Table 1 ijms-24-12966-t001:** Clinical characteristics of all patients with cognitive impairment (*N* = 145) and of patients with AD (*N* = 82), MCI (*N* = 28) and MCI (NOT AD) diagnosis (*N* = 35).

Characteristic	Category/Unit	Cognitive Impairment	AD	MCI (AD)	MCI (NOT AD)	*p*
Sex	Male, *N* (%)	68 (46.9)	35 (42.7)	9 (32.1)	24 (68.6)	0.008 ^a^
	Female, *N* (%)	77 (53.1)	47 (57.3)	19 (67.9)	11 (31.4)	
Age	Years, median (25–75%)	76 (72–80)	77 (73.75–80)	75 (73–79.75)	74 (67–78)	0.039 ^b^
Education	Years, median (25–75%)	12 (10–14.25) [3]	11.5 (8.5–12) [2]	12 (12–16) [1]	12 (11–15)	0.022 ^b^
Weight	kg, median (25–75%)	70 (59–79.75) [25]	67 (55.5–78) [17]	67 (56–78) [9]	75.5 (64.75–85) [3]	0.011 ^b^
BMI	kg/m^2^, median (25–75%)	24.49 (20.99–27.01) [26]	24.22 (20.23–26.61) [16]	23.01 (20.45–27.54) [7]	26.05 (23.74–28.57) [3]	0.008 ^b^
APOE status	*APOE4* carriers, N (%)	66 (45.5)	43 (52.4)	16 (57.1)	7 (20)	0.005 ^a^
MMSE	Score, median (25–75%)	24 (19.75–26) [27]	20 (16–23) [19]	27 (25.5–27) [7]	26 (25–27) [9]	<0.001 ^b^
Aβ_42_	pg/mL, median (25–75%)	740 (572.5–997.5)	678.5 (542–771)	648.5 (514.75–828)	1305 (1119–1496)	<0.001 ^b^
Aβ_42/40_ ratio	Median (25–75%)	0.06 (0.05–0.08) [6]	0.06 (0.04–0.06) [2]	0.05 (0.04–0.06) [2]	0.12 (0.09–0.14) [2]	<0.001 ^b^
Total tau	pg/mL, median (25–75%)	567 (418.5–879.5)	771 (537–989)	594 (463.25–910.75)	311 (239–404)	<0.001 ^b^
p-tau_181_	pg/mL, median (25–75%)	87 (64.5–116.5)	98 (81–125.25)	89 (77.25–128.5)	50 (39–62)	0.008 ^b^

AD: Alzheimer’s disease; BMI: body mass index; MCI: mild cognitive impairment; MMSE: The Mini-mental State Exam. ^a^ Fisher’s exact test; ^b^ Kruskal–Wallis test.

**Table 2 ijms-24-12966-t002:** Comparison of genotype frequencies among patients with different types of cognitive impairment.

Gene	SNP	Genotype	AD *N* (%)	MCI *N* (%)	MCI (NOT AD) *N* (%)	*p*
*SORCS1*	rs1358030	CC	8 (9.8)	4 (14.3)	5 (14.3)	0.467
		CT	33 (40.2)	14 (50)	18 (51.4)	
		TT	41 (50)	10 (35.7)	12 (34.3)	
		CT + TT	74 (90.2)	24 (85.7)	30 (85.8)	P_dom_ = 0.639
	rs1416406	TT	9 (11)	4 (14.3)	2 (5.7)	0.847
		TC	30 (36.6)	9 (32.1)	13 (37.1)	
		CC	43 (52.4)	15 (53.6)	20 (57.1)	
		TC + CC	73 (89)	24 (85.7)	33 (94.3)	P_dom_ = 0.528
*BCHE*	rs1803274	CC	52 (63.4)	19 (67.9)	26 (74.3)	0.741
		CT	28 (34.1)	8 (28.6)	8 (22.9)	
		TT	2 (2.4)	1 (3.6)	1 (2.9)	
		CT + TT	30 (36.5)	8 (28.6)	9 (25.7)	P_dom_ = 1
	rs1799807	TT	78 (95.1)	27 (96.4)	34 (97.1)	1
		TC	4 (4.9)	1 (3.6)	1 (2.9)	
		CC	0	0	0	
		TC + CC	4 (4.9)	1 (3.6)	1 (2.9)	P_dom_ = 1
*TOMM40*	rs2075650	AA	42 (51.2)	12 (42.9)	26 (74.3)	0.082
		AG	34 (41.5)	13 (46.4)	7 (20)	
		GG	6 (7.3)	3 (10.7)	2 (5.7)	
		AG + GG	40 (49.8)	16 (57.1)	9 (25.7)	**P_dom_** = **0.025**
	rs157581	TT	28 (34.1)	7 (25)	23 (65.7)	**0.005**
		TC	44 (53.7)	17 (60.7)	8 (22.9)	
		CC	10 (12.2)	4 (14.3)	4 (11.4)	
		TC + CC	54 (65.9)	21 (75)	12 (34.3)	**P_dom_** = **0.001**
*ACE*	rs1800764	CC	21 (25.6)	4 (14.3)	4 (11.4)	0.317
		CT	38 (46.3)	15 (53.6)	16 (45.7)	
		TT	23 (28)	9 (32.1)	15 (42.9)	
		CT + TT	61 (74.4)	24 (85.7)	31 (88.6)	P_dom_ = 0.194
	rs4343	GG	27 (32.9)	7 (25)	8 (22.9)	0.360
		GA	39 (47.6)	13 (46.4)	14 (40)	
		AA	16 (19.5)	8 (28.6)	13 (37.1)	
		GA + AA	55 (67.1)	21 (75)	27 (77.1)	P_dom_ = 0.506
*IL6 R*	rs2228145	AA	31 (37.8)	15 (53.6)	14 (40)	0.683
		AC	43 (52.3)	11 (39.3)	17 (48.6)	
		CC	8 (9.8)	2 (7.1)	4 (11.4)	
		AC + CC	51 (62.2)	13 (46.4)	21 (60)	P_dom_ = 0.342

**Table 3 ijms-24-12966-t003:** Association of investigated polymorphisms with cerebrospinal fluid biomarkers among all patients with cognitive impairment.

SNP	Genotype	Aβ_42_ (pg/mL)	*p*	Aβ_42_/_40_ Ratio	*p*	Total tau (pg/mL)	*p*	p-tau (pg/mL)	*p*
*SORCS1* rs1358030	CC	774 (560.5–1048)	0.467	0.05 (0.04–0.08)	0.792	769 (463.5–1240.5)	0.301	95 (62.5–152)	0.767
	CT	740 (592.5–1114.5)		0.06 (0.04–0.09)		556 (379.5–849.5)		84 (62.5–116)	
	TT	725 (538–877)		0.06 (0.05–0.08)		567 (427–878)		87 (65–114)	
	CT + TT	735.5 (576.5–981.5)	P_dom_ = 0.701	0.06 (0.05–0.08)	P_dom_ = 0.533	557.5 (417.75–861.75)	P_dom_ = 0.148	86.5 (64.25–114.75)	P_dom_ = 0.499
*SORCS1* rs1416406	TT	702 (570–916)	0.944	0.06 (0.04–0.07)	0.952	719 (461–894)	0.175	90 (66–116)	0.271
	TC	744 (576.5–1066.5)		0.06 (0.05–0.08)		522.5 (411.75–774.25)		81 (60.5–104.5)	
	CC	739 (566.5–1071.25)		0.06 (0.04–0.09)		615 (421.25–989)		91 (66.5–126.5)	
	TC + CC	743.5 (573.75–1071.25)	P_dom_ = 0.745	0.06 (0.05–0.08)	P_dom_ = 0.877	557.5 (415.25–878.75)	P_dom_ = 0.745	85 (63.75–117.5)	P_dom_ = 0.768
*BCHE* rs1803274	CC	758.5 (567.75–1075.75)	0.487	0.06 (0.05–0.08)	0.271	552.5 (403–840)	0.306	81 (60.75–111)	0.136
	CT	678.5 (566.5–893.5)		0.06 (0.04–0.07)		612.5 (440.25–1052.75)		97.5 (67.75–140)	
	CT + TT	688 (563–851.5)	P_dom_ = 0.317	0.06 (0.04–0.07)	P_dom_ = 0.275	617 (437.5–1035)	P_dom_ = 0.201	96 (69–135.5)	P_dom_ = 0.062
*BCHE* rs1799807	TT	743 (570–1005)	1	0.06 (0.04–0.08)	0.471	567 (420–878)	0.929	86 (65–116)	0.897
	TC	697 (661.25–958)		0.07 (0.06–0.08)		564.5 (329.5–1082)		89 (49.5–135.25)	
	TC + CC	697 (661.25–958)	P_dom_ = 1	0.07 (0.06–0.08)	P_dom_ = 0.471	564.5 (329.5–1082)	P_dom_ = 0.929	89 (49.5–135.25)	P_dom_ = 0.897
*TOMM40* rs2075650	AA	765 (591.75–1155.75)	0.178	0.06 (0.04–0.09)	0.170	552.5 (403–874.5)	0.319	81 (58.5–110)	0.203
	AG	698 (579.5–878)		0.06 (0.05–0.07)		592.5 (459.75–859.25)		93.5 (69.5–120.25)	
	GG	613 (513–877)		0.05 (0.03–0.06)		821 (385–1183)		114 (59–151)	
	AG + GG	697 (558.5–871.5)	P_dom_ = 0.095	0.06 (0.05–0.07)	P_dom_ = 0.219	613 (458.5–940)	P_dom_ = 0.219	95 (69–124)	P_dom_ = 0.099
*TOMM40* rs157581	TT	799.5 (593.25–1234.25)	0.095	0.06 (0.04–0.11)	0.239	543 (319–777.25)	0.099	81 (56.75–109)	0.106
	TC	720 (565.5–885.5)		0.06 (0.05–0.07)		613 (470–886)		95 (74–118)	
	CC	682.5 (520.75–815.5)		0.06 (0.04–0.09)		768 (374.75–1070.25)		97.5 (57.5–131.75)	
	TC + CC	711 (561–877)	P_dom_ = 0.033	0.06 (0.05–0.07)	P_dom_ = 0.091	617 (468–911)	P_dom_ = 0.032	95 (73–123)	P_dom_ = 0.034
*ACE* rs1800764	CC	658 (542.5–899.5)	0.341	0.06 (0.04–0.07)	0.432	617 (459.5–888)	0.641	97 (73.5–124)	0.469
	CT	759 (613.5–948.5)		0.06 (0.05–0.08)		549 (448.5–879)		84 (66.5–115)	
	TT	743 (575–1291)		0.06 (0.04–0.11)		567 (403–881)		81 (58–114)	
	CT + TT	747.5 (596.75–1071.75)	P_dom_ = 0.153	0.06 (0.05–0.08)	P_dom_ = 0.204	554 (412.5–876.75)	P_dom_ = 0.416	82 (63.25–114)	P_dom_ = 0.256
*ACE* rs4343	GG	712 (558.5–892.75)	0.592	0.06 (0.05–0.07)	0.979	594.5 (343.75–859.25)	0.594	95 (59.75–123.25)	0.732
	GA	751 (613.75–926)		0.06 (0.05–0.08)		560 (467.75–894.5)		86.5 (67.75–115.25)	
	AA	747 (545–1326.5)		0.06 (0.04–0.11)		515 (392.5–913)		81 (57.5–122.5)	
	GA + AA	747 (585–1071)	P_dom_ = 0.353	0.06 (0.05–0.08)	P_dom_ = 0.845	549 (427–896)	P_dom_ = 0.819	83 (65–116)	P_dom_ = 0.642
*IL6 R* rs2228145	AA	718.5 (540–938.5)	0.204	0.06 (0.04–0.08)	0.984	557.5 (428.25–910.75)	0.771	85.5 (68–122)	0.796
	AC	760 (602–1072)		0.06 (0.05–0.08)		571 (410–855)		87 (65–111)	
	CC	636 (529–977.5)		0.06 (0.04–0.10)		688.5 (297.75–1075)		82 (40.75–134.5)	
	AC + CC	747 (592.5–1071.5)	P_dom_ = 0.343	0.06 (0.05–0.08)	P_dom_ = 0.856	571 (407–875)	P_dom_ = 0.757	87 (63.5–114.5)	P_dom_ = 0.518

## Data Availability

All the data are presented within the article and in the Appendix A. Any additional information is available on request from the corresponding author.

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
