# Peer review of "The Association of Selected GWAS Reported AD Risk Loci with CSF Biomarker Levels and Cognitive Decline in Slovenian Patients"

_ijms, 2023, doi:10.3390/ijms241612966_

Round 1

Reviewer 1 Report

This is an interesting study. I have a few suggestions which may improve it.

1. In the Methods section, I would add a workflow of experiments.

2. Also in the methods I would describe the CSF Tau and amyloid-beta analysis a little bit more in detail. 

3. Were there any similar studies with GWAS data and plasma amyloid beta levels? 

4. Are the genes (tomm40, SORCS, BCHE, IL6R, and ACE) have some common pathway through AD onset, or do they act independently from each other?

5. Were TOMM40, ACE, and IL6 variants associated with any biomarker changes and cognitive impairment? The biomarker data of these variants may be added either in the manuscript or in a supplement file. 

7. Were these variants studied (in terms of AD) in other populations? (such as in other European countries or in Asian/American patients?) The authors may mention this in the discussion briefly. 

6. It would be nice to repeat this study with a larger sample size. 

Author Response

On behalf of all co-authors, we submit the reply to the reviewers below, where all the questions and comments of the Reviewers were taken into consideration and the revised manuscript incorporates all due revisions and explanatory comments. 

Reviewer 2 Report

Authors present the interesting correlations between CSG biomarkers and genetics from GEAS.

However, authors misrepresented the data on Table 1. Clinical characteristics of all patients with cognitive impairment or need to carefully review all the values in all categories ONE by ONE. 

Authors did not check Table 1 carefully.

If the values in the Table 1 is correct, the results cannot be trusted.

Author Response

On behalf of all co-authors we submit the reply to the reviewers below, where all the questions and comments of the Reviewers were taken into consideration and the revised manuscript incorporates all due revisions and explanatory comments. 

Round 2

Reviewer 1 Report

The authors fulfilled my suggestions.